

# Influence of tannic acid concentration on the physicochemical characteristics of saliva of spider monkeys (*Ateles geoffroyi*)

Carlos Eduardo Ramírez-Torres[1], Fabiola Carolina Espinosa-Gómez[2], Jorge E. Morales-Mávil[1], J. Eduardo Reynoso-Cruz[1], Matthias Laska[3] and Laura Teresa Hernández-Salazar[1]

[1] Instituto de Neuroetología, Universidad Veracruzana, Xalapa, Veracruz, México
[2] Facultad de Medicina Veterinaria y Zootecnia, Universidad Popular Autonóma del Estado de Puebla (UPAEP), Puebla, Puebla, México
[3] Department of Physics, Chemistry and Biology, Linköping University, Linköping, Sweden, Sweden

## ABSTRACT

Tannins are a chemical defense mechanism of plants consumed by herbivores. Variations in salivary physicochemical characteristics such as pH, total protein concentration (TP), and presence of proline-rich proteins (PRPs) in animals have been reported as a mechanism to protect the oral cavity when consuming food with variations in pH and tannins. Variations in salivary physiochemistry as adaptations for consuming tannin-rich foods have been found in omnivorous and folivorous primates, but have not yet been reported in frugivorous species such as spider monkeys. We therefore assessed changes in pH using test strips, TP concentration by measuring absorbance at 595 nm in a spectrophotometer and salivary PRPs using the SDS-PAGE electrophoresis technique in the saliva of nine captive spider monkeys in response to the consumption of solutions with different concentrations of tannic acid. The results showed variations in pH, TP concentration and the presence and variation of possible salivary PRPs associated with tannic acid concentration. These findings suggest that spider monkeys may tailor their salivary physicochemical characteristics in response to the ingestion of potentially toxic compounds.

## INTRODUCTION

Plant secondary metabolites such as tannins are known to serve as chemical defense mechanisms against predation by herbivores (*War et al., 2012*). They are present in different parts of plants such as fruits, bark, leaves, and seeds (*Clauss, 2003*). Depending on the plant part, the concentration of these compounds may vary. For example, there is usually a higher concentration of tannins in leaves than in fruits (*Cork & Foley, 1991*; *Espinosa-Gómez et al., 2013*). There are also variations in the concentration of tannins depending on the ripeness of the plant part, with ripe fruit reported to have lower concentrations compared to unripe fruit (*Kreuger & Potter, 1994*; *Bashir & Abu-Goukh, 2003*; *Del Bubba et al., 2009*; *Belwal et al., 2019*). In addition to the differences in the

Corresponding authors
Fabiola Carolina Espinosa-Gómez, fabiespinosa.mvz@gmail.com
Laura Teresa Hernández-Salazar, herlatss@gmail.com

concentration of tannins in ripe and unripe fruits, there are also changes in the concentration of organic acids, which are higher in unripe than in ripe fruits since they are precursors to the carbohydrates present in the ripe fruits (*Nelson et al., 2000*; *Bourgaud et al., 2001*; *Espinosa-Gómez et al., 2018*; *Batista-Silva et al., 2018*).

At low concentrations tannins can act as antioxidants (*Crozier, Jaganath & Clifford, 2009*; *Gibbins & Carpenter, 2013*), while at high concentrations they can reduce nutrient absorption (*Bernays, Driver & Bilgener, 1989*; *Harborne, 1991*; *Dixon, Xie & Sharma, 2005*) due to their ability to precipitate proteins and inhibit gastrointestinal enzymes, resulting in reduced digestibility of the ingested plant parts (*Ryan, 1979*; *Waterman et al., 1980*; *Bennick, 2002*; *Dixon, Xie & Sharma, 2005*). This reduced digestibility can ultimately affect the growth rate and development of individuals consuming them (*Robbins et al., 1987*; *Shimada, 2006*). One tannin that has been shown to be present in different fruits is tannic acid (*Chung et al., 1998*; *Gülçin et al., 2010*; *Ghosh, 2015*), which is frequently used to assess effects of tannins in animals that consume them (*Becker & Makkar, 1999*; *Laska et al., 2000*; *Park et al., 2002*; *Alonso-Díaz et al., 2012*).

Animals that include plants in their diet are able to detect the presence of tannins, as these polyphenols modify the organoleptic properties of the food by generating a taste sensation described as bitter/astringent (*Schobel et al., 2014*; *Lamy et al., 2016*). At the moment of experiencing this taste sensation, animals may respond by avoiding the consumption of plant parts presenting these characteristics (*Hagerman, 1992*; *Pavagadhi & Swarup, 2020*). For many herbivores it is impossible to avoid the consumption of a plant part, and many have evolved strategies that allow them to feed on the plants despite the presence of these types of compounds (*Boze et al., 2010*; *Barbehenn & Peter Constabel, 2011*; *Espinosa-Gómez et al., 2018*). This has been observed to occur in some primate species such as diademed sifakas (*Propithecus diadema*) (*Thurau, Rahajanirina & Irwin, 2021*) and Peruvian spider monkeys (*Ateles chamek*) (*Felton et al., 2009*), which continue to consume tannin-containing foods, so it is likely that these have evolved some physiological mechanisms to cope with dietary tannins (*Felton et al., 2009*; *Thurau, Rahajanirina & Irwin, 2021*).

It has been proposed that herbivores are able to respond to the presence of tannins and dietary acids by changing their oral environment, modifying salivary characteristics during mastication to counteract the acidity and secondary metabolites of foods through the modification of salivary pH and the production of salivary proteins (*Beal, 1991*; *Llena-Puy, 2006*; *Neyraud, Bult & Dransfield, 2009*; *Canon et al., 2010*; *Boze et al., 2010*; *Lavy et al., 2012*). Changes in the salivary protein concentration associated with the diet have been described. Primates that include a higher proportion of fiber in their diet like macaques, gorillas, and chimpanzees have higher concentrations of total protein (TP) compared to others that consume less fiber, such as humans (*Thamadilok et al., 2020*).

Tannin-binding salivary proteins (TBSPs) form tannin-protein complexes that prevent tannins from binding to other proteins in the intestine such as digestive enzymes and thereby affect nutritional intake (*Hagerman & Butler, 1980*). The two major groups of these proteins are recognized as proline-rich proteins (PRPs) and histatins (*Hagerman et al., 1998*; *Naurato et al., 1999*; *Shimada, 2006*; *Canon et al., 2010*). PRPs are

considered to be the main line of defense against dietary tannins, as they are found in higher concentrations in the saliva compared to histatins (*Hagerman et al., 1998*; *Shimada, 2006*). PRPs have been reported in the saliva of non-human primates with different types of diets, such as species with omnivorous diets like baboons (*Papio hamadryas*), macaques (*Macaca fascicularis*, *Macaca mulata* and *Macaca arctoides*) and the vervet monkey (*Cercopithecus aethiops*; *Jacobsen & Arneberg, 1976*; *Oppenheim, Kousvelari & Troxler, 1979*; *Schlesinger, Hay & Levine, 1989*; *Mau et al., 2011*). They have also been reported in species with predominantly folivorous diets such as mantled howler monkeys (*Alouatta palliata mexicana*; *Espinosa Gómez et al., 2015*) and black howler monkeys (*Alouatta pigra*; *Espinosa-Gómez et al., 2018*). However, the presence of PRPs has not been studied in primates with a predominantly frugivorous diet, whose potential responsiveness might be different compared to that of omnivorous or folivorous species, since, as mentioned previously, fruits have a lower concentration of tannins compared to other plant parts (*Cork & Foley, 1991*; *Batista-Silva et al., 2018*; *Espinosa-Gómez et al., 2018*).

Spider monkeys are mainly frugivorous (*Klein & Klein, 1977*) and spend between 75% to 90% of their foraging time-consuming ripe fruits (*Klein & Klein, 1977*; *Cant, 1990*; *Wallace, 2005*; *González-Zamora et al., 2009*). They inhabit neotropical forests, where seasonal differences in precipitation lead to changes in fruit availability (*Di Fiore & Link, 2008*). These changes in the availability of fruits force the spider monkeys to modify their diet to at least seasonally include other plant parts (*Chapman & Chapman, 1990*; *Wallace, 2005*; *Felton et al., 2008*; *Felton et al., 2009*). This exposes the spider monkey to different concentrations of plant secondary metabolites and, consequently, to the potential effects on digestion and health associated with their consumption (*Chapman, 1987*; *Di Fiore & Link, 2008*; *Felton et al., 2008*; *Wallace, 2005*; *Felton et al., 2009*). As a result, spider monkeys might require strategies to cope with the presence of different concentrations of plant secondary metabolites such as tannins, and these may include the modification of salivary physicochemical characteristics such as pH and proteins.

It was therefore the aim of this study assess the physicochemical characteristics (pH, TP concentration and presence of PRPs) in saliva of black-handed spider monkeys (*A. geoffroyi*) when consuming solutions with different concentrations of tannic acid.

## MATERIALS AND METHODS

### Ethics

The experiments reported here comply with the Guideline for the care and use of mammals in neuroscience and behavioral research (*National Research Council, 2003*), the *American Society of Primatologists' Principles for the Ethical Treatment of Primates*, and with Mexican laws (NOM-062-ZOO-1999 and NOM-051-ZOO-1995). The protocol was approved by the Ethics Council of the Ministry of Environment and Natural Resources (SEMARNAT; official permits number 09/GS-2132/05/10).

### Study individuals and housing conditions

We worked with nine adult black-handed spider monkeys (*A. geoffroyi*), five males and four females. The animals were kept under human care, living in enclosures (4 × 4 × 4 m)

connected by sliding doors that allow them to interact and to be temporarily separated for individual testing. All individuals were exposed to natural environmental conditions of temperature, relative humidity and light-dark cycles. The enclosures were located at the natural reserve Hilda Avila de O'Farrill managed by the Instituto de Neuroetología of the Universidad Veracruzana near the town of Catemaco Veracruz, in the south-east of Mexico. The monkeys were not deprived of food, but all of the experiments were conducted before feeding time to ensure that the animals were motivated by the substances that acted as food rewards. The diet of the spider monkeys was based on cultivated fruits and vegetables, and changed seasonally according to availability.

## Experimental design

The nine monkeys were divided into two groups: Experimental and control.
The experimental group consisted of five adult spider monkeys (three males and two females), while the control group consisted of four adult individuals (two males and two females). We presented the individuals with bottles containing 100 ml of each solution with different tannin concentrations with sucrose for 1 min. Individuals in the experimental group consumed a solution of 30 mM of sucrose with different concentrations of tannic acid (0.01, 0.05, 0.1, 0.5 and 1 mM), while individuals in the control group always consumed only 30 mM sucrose without tannic acid. The different concentrations of tannic acid were administered to the individuals in the experimental group in ascending order, starting with the solution of only 30 mM of sucrose (control solution) and ending with the highest concentration of tannic acid (1 mM). Figure 1 shows the experimental design of the study.

The sucrose concentration used here has been shown to be above the taste preference threshold of spider monkeys, making it sufficiently attractive to induce consumption and yet low enough not to cause masking effects to other substances (*Laska et al., 1996*; *Laska et al., 2000*). The concentrations of tannic acid used here were above the taste preference threshold of *A. geoffroyi* (*Laska et al., 2000*), which ensured that the individuals were able to perceive them. The administration and sampling were carried out from 8:00 to 10:00 am. The Animals had a period without access to the tannic acid solution, and during this time the individuals were only presented with a 30 mM sucrose solution. The sucrose (CAS# 57-50-1) was obtained from Merck and the tannic acid (CAS# 1401-55-4) was obtained from Meyer.

## Saliva collection

The method employed here is a variation of the nylon swab method, which has been used to extract salivary samples from other non-human primates (*Smiley Evans et al., 2015*). The spider monkeys were trained to chew a swab (SalivaBio Children's Swab, Salimetrics 5001.08, SalivaBio, State College, PA, USA) that absorbs saliva. The swabs were 13 cm long and were adjusted to a size of 4.3 cm, so that the monkeys could fit the whole swab into their oral cavity. To encourage the monkeys to chew the swab, they were soaked with 0.5 ml of corn syrup. One end of the swab was tied with a 30 cm long cotton string that was held in place while the monkeys chewed the swab. The monkeys were allowed to chew the

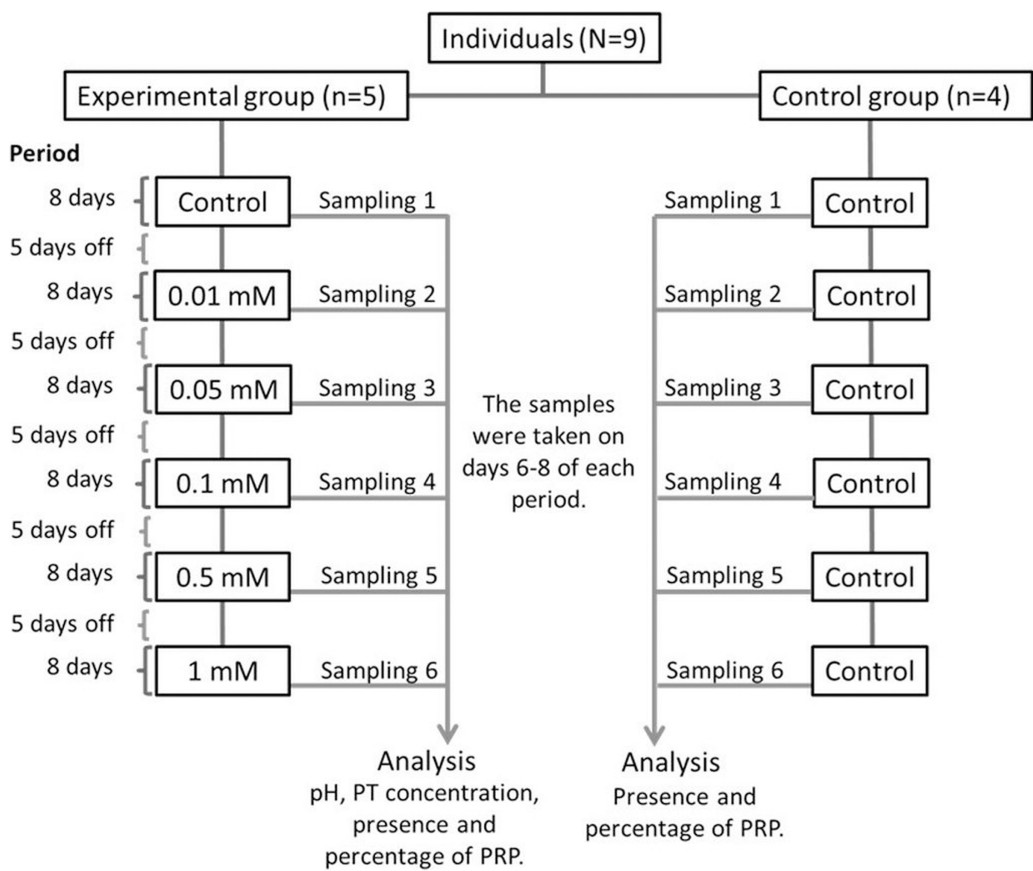

**Figure 1 Experimental design.** All of the solutions were administered to both the experimental and control groups for a period of 8 days and the administration of the solutions were always given in the same order to the individuals. Between each period of administration of a solution the spider monkeys received five days off where the monkeys received only a solution of water with 30 mM of sucrose so that the data obtained would be in response to each solution and not to the accumulated response. Salivary samples from both groups were collected from day 6 to 8 of each period when a solution was administered. Samples from individuals in the experimental group were used to determine pH, TP concentration, presence and percentage of PRPs, while those from the control group were only used to determine the presence and percentage of PRPs.

swab for 60 s, which ensured that the swab became soaked with saliva. Once the chewing time was completed, the string was gently pulled to retrieve the swab. The swab was placed in a 3 ml syringe to compress it and recover the saliva in a microtube (1.5 ml), then the saliva was stored and frozen in liquid nitrogen (−196 °C). All procedures were carried out under hygienic conditions to avoid contamination of the samples.

## Determination of salivary acidity/alkalinity (pH)

The salivary pH value was determined using test strips (HICARER®). For this purpose, individuals were trained to lick the strips on command to impregnate them with enough saliva for the determination of pH by color comparison with the standards.

## Processing of salivary samples

We administered the solutions to the experimental and control groups for eight days. Salivary samples from both groups were collected from days 6 to 8 of each period when a solution was presented. The saliva from the three collection days (6 to 8) was pooled for each individual. This procedure was performed with each of the concentrations of the experimental group and the equivalent days of the control group. The samples were removed from liquid nitrogen and thawed in an ice bath. Saliva aliquots were thawed, cells and debris were removed by centrifugation at 16,000 g for 10 min at 4 °C, and the supernatant was captured. The soluble fraction (supernatant) was preserved and a protease inhibitor cocktail was added in a 500:1 ratio; the samples were stored at −80 °C until further analysis. All of the saliva analyses were carried out at the Veterinary School of the Universidad Popular Autónoma del Estado de Puebla (UPAEP).

## Determination of the total concentration of salivary proteins (TP concentration)

TP concentration was measured using the Bradford method, using a spectrophotometer measuring the absorbance of proteins at 595 nm (*Bradford, 1976*). This method has been used previously for identification of TP concentration in the saliva in other non-human primates (*Espinosa-Gómez et al., 2018*).

## Determination of the presence of proline-rich proteins (PRPs)

The presence of PRPs in monkeys' saliva was determined according to the technique proposed by *Beeley et al. (1996)*, which consists of the identification of protein bands in one-dimensional SDS-PAGE electrophoresis gel. We separated salivary proteins using 12% one-dimensional sodium dodecyl sulfate-polyacrylamide gel electrophoresis (SDS-PAGE) with running buffer (0.03 M Tris, 0.144 M glycine, 0.1% (w/s) SDS, pH 8.3), following the procedure of *Laemmli (1970)*. The 1D-SDS PAGE was run with the maximum volume (~30 mg of total protein) of the saliva samples into the wells of the electrophoresis gels. The sample to be loaded on the gel consisted of a mixture of saliva with SDS loading buffer 2:1 (0.125 M Tris-HCL pH 6.8, 2% SDS, 5% 2-mercaptoethanol, 20% glycerol with traces of bromophenol blue). We then incubated the mixture in a boiling water bath (5 min) to denature the proteins. A current of 120 V was applied for a period of 90 min. After running the gels, they were placed in two different protein fixatives one after the other with 1 h in each fixative. The first was 26% ethanol, 14% formaldehyde, and 60% distilled water; the second was 50% methanol, 12% acetic acid, and 38% distilled water. The next step was to dye the gels in a bath of Coomassie-R250 for 12 h and after that they were destained with a solution of 10% acetic acid. This technique allowed us to observe the presence of pink or violet bands, which indicate the presence of PRPs, frequently in the range of 15 to 30 kDa (*Beeley et al., 1996*).

We used 5 μl of the molecular weight marker BLUltra Prestained Protein Ladder (BIO-HELIX®), which was loaded in the first lane. Samples from individuals in the experimental group were loaded in the next five lanes and samples from the control group were loaded in the last four lanes.

### Densitometry analysis of protein bands identified as proline-rich proteins (PRPs) in 1D-SDS-PAGE gels

To calculate the percentage of possible PRPs (%PRPs) the electrophoresis gels were scanned at 1,200 dpi quality with an HP Digital Sender Flow 8,500 fn2 scanner. The images from the gels of the control and the experimental group were processed with a densitometry analysis using the IMAGEJ software (*Tiago & Wayne, 2012*). During this analysis the software quantifies the megapixels of each band to determine their size and density. The densitometry analysis estimates how much (in percentage) the PRP bands represent in relation to all of the bands in the sample. The software identified between nine to 10 bands of salivary proteins. The band of possible PRPs was band seven when nine bands were recorded or band eight when ten bands were recorded.

### Statistical analysis

Means and standard errors (S.E.) were calculated using the individual results from the salivary physicochemical characteristics of the spider monkeys (pH, TP concentration and %PRPs). Those individual results of the physicochemical characteristics obtained from the six samplings of the experimental group were compared using a Friedman test and Nemenyi's *post hoc* tests to determine differences between the control solution and the five concentrations of tannic acid. To determine if there was a relationship between the tannic acid concentration and the physicochemical characteristics, Pearson correlation tests were performed. To test the effects of the consumption of different concentrations of tannic acid on the physicochemical characteristics considered in this study, linear mixed models (LMM) were performed. All analyses were performed in R statistical software. (version 4.0.0; *R Core Team, 2020*)

## RESULTS

### pH

The Friedman test showed that there were statistically significant differences between the salivary pH across the solutions with different concentrations of tannic acid that were presented to the individuals ($x^2$ (5) = 17.857, $p$ = 0.003, w = 0.714). The Nemenyi test identified differences in the pH between the solutions of 1 mM and the control solution ($p$ = 0.007), and between the solutions of 1 and 0.1 mM ($p$ = 0.022). The lowest mean value of the saliva pH (8 ± 0.1) was obtained when the spider monkeys consumed the control solution and the highest value was obtained when the spider monkeys consumed the solution with the highest concentration of tannic acid of 1 mM (8.77 ± 0.08) (Fig. 2A). A statistically significant correlation was found between the consumption of the solution with tannic acid concentration and the pH values of the spider monkeys' saliva (r = 0.694, $p$ < 0.001; Fig. 2B).

### Total protein (TP)

A Friedman's test indicated significant differences ($x^2$ (5) = 14.422, $p$ = 0.013, w = 0.577) between the mean values (±S.E.) of TP obtained during the administration of the control solution and the solutions with different concentrations of tannic acid (Fig. 3A).

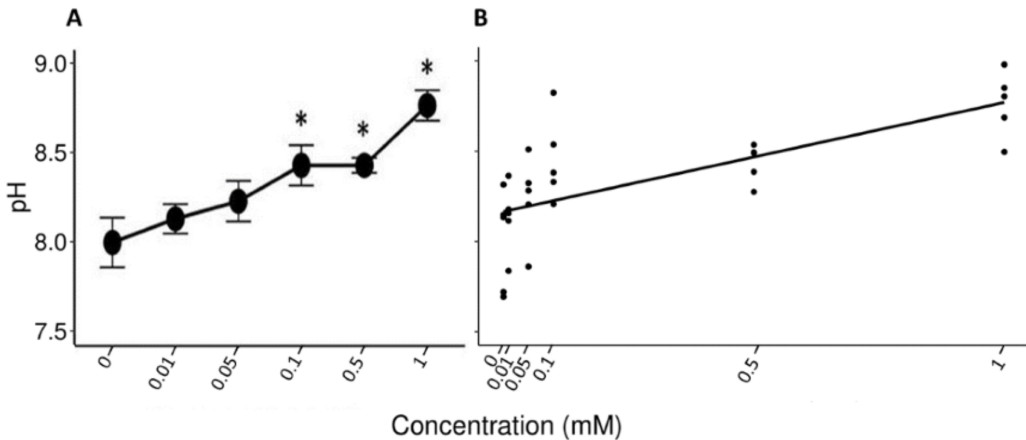

**Figure 2 Variation of salivary pH to different concentrations of tannic acid.** (A) Mean values ± S.E of salivary pH as a function of the solutions with different concentrations of tannic acid. The asterisks indicate the solutions that showed significant differences with respect to the samples of the experimental group when given the control solution. (B) Correlation between different concentrations of tannic acid and the salivary pH. The pH values obtained with the concentrations of tannic acid, so the data for the 0.01 mM concentrations are very close to each other on the graph.

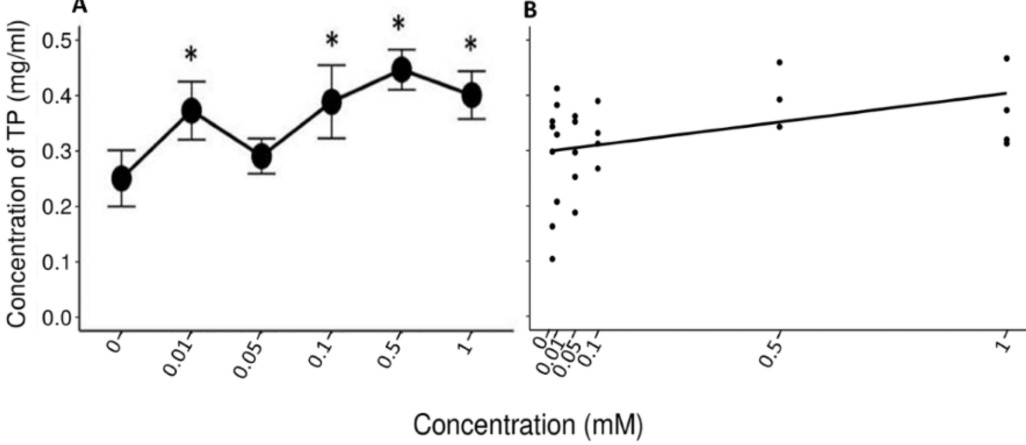

**Figure 3 Variation of TP concentration to different concentrations of tannic acid.** (A) Mean values ± S.E of the TP concentration as a function of the solutions with different concentrations of tannic acid. The asterisks indicate the solutions that showed significant differences with respect to the samples of the experimental group when given the control solution. (B) Correlation between different concentrations of tannic acid and the TP concentration. The TP values obtained with the concentrations of tannic acid are on a real scale, so the data for the 0 and 0.01 mM concentrations are very close to each other on the graph.

The Nemenyi tests showed only significant differences between the solution of 0.5 mM of tannic acid and the control solution with respect to the other solutions ($p = 0.028$). The Pearson correlation between the consumed solutions with different concentrations of tannic acid and the TP concentration in the saliva of the spider monkeys was positive but relatively small (r= 0. 335, $p = 0.034$, Fig. 3B).

## Percentage of proteins rich in prolin (%PRP)

The data of the experimental group indicated an increase in the %PRPs present in the saliva of spider monkeys as the concentration of tannic acid increased. The mean value of %PRPs during the first sampling (control solution) was 15.90%, while the mean value during the sixth sampling (1 mM) was 47.34%. The mean values of the %PRPs in saliva of the individuals of the control group remained in a range of 16.66% to 30.15% (Fig. 4).

The Friedman test indicated a statistically significant difference ($x^2$ (5) = 14.371, $p = 0.013$, w = 0.575; Fig. 5A) and Nemenyi's test showed that this difference was between the control solution and all solutions containing tannic acid ($p = 0.002$). The concentration of tannic acid in the solutions and the %PRPs in saliva showed a significant positive correlation (r = 0.563, $p < 0.001$; Fig. 5B).

The LMM indicate the $r^2$ values and the confidence intervals at 95% (CI) were calculated using the r2beta package. In all models, the concentration of tannic acid was set as a fixed effect and the IDs of the individual monkeys in the study as a random effect. The results of the model of the pH of the saliva indicated a significant effect of concentration of tannic acid in the consumed solution (t = 2.483, $p = 0.042$, CI = 0.01–0.456). As the concentration of tannic acid increases, the pH becomes more alkaline. However, the random effect indicated that the variations in the pH show differences between individuals (SD = 6.94). For the TP concentration the results indicate a significant effect of concentration of tannic acid in the consumed solution (t = 2.655, $p = 0.033$, CI = 0.136–0.622); so, as the concentration of tannic acid increases, the TP concentration also increases, while the random effects show minimal variations among the individuals (SD = 0.06). In the case of %PRPs, the model showed a significant effect of concentration of tannic acid (t = 5.075, $p = 0.001$, CI = 0.242–0.694), indicating that the consumption of higher concentrations of tannic acid produces a higher %PRPs. This result was very stable across individuals since the random effect indicated only minor variation among the monkeys (SD < 0.001). In relation to the $r^2$ values obtained from the analysis, the value for the %PRPs ($r^2 = 0.479$) was the highest, while the lowest values were obtained for TP concentration ($r^2 = 0.377$), and pH ($r^2 = 0.18$).

## DISCUSSION

We found that spider monkeys (*A. geoffroyi*) are able to modify the physicochemical characteristics of their saliva (pH and protein production) in response to the consumption of different concentrations of tannic acid, which indicates a physiological adaptation to deal with tannins and organic acids present in the food they eat.

### pH

The results of this study showed that under baseline conditions (control) the salivary pH of spider monkeys is slightly alkaline, similar to the mean value of pH reported for black howler monkeys (seven to eight in the wild and eight in the zoo; *Espinosa-Gómez et al., 2018*). We found variations in the values of pH throughout the sampling periods in the experimental group. The highest pH values, ranging from 8.43–8.77 (more alkaline saliva), were recorded when individuals consumed the highest concentrations of tannic acid

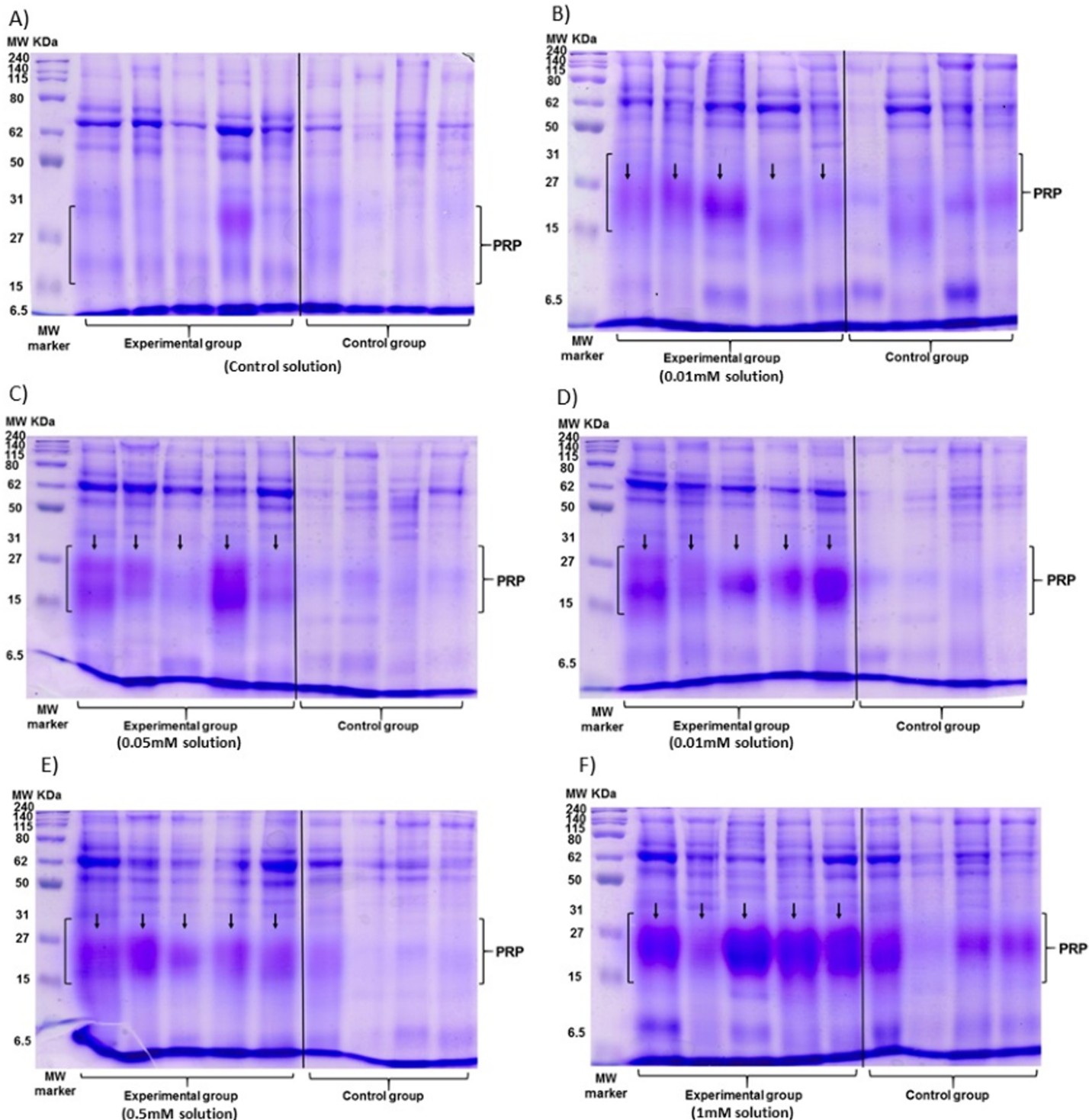

**Figure 4 Visual representation of SDS-PAGE gels of the samples of saliva of the spider monkeys.** The left side of all images belongs to the experimental group while the right side belongs to the control group. The letter indicates the concentration of tannic acid consumed by the experimental group. (A) The control solution, (B) the 0.01 mM solution, (C) the 0.05 mM solution, (D) the 0.1 mM solution, (E) the 0.5 mM solution, and (F) the 1 mM solution. The individuals in the control group always consumed the solution with 30 mM of sucrose. vertical black line.

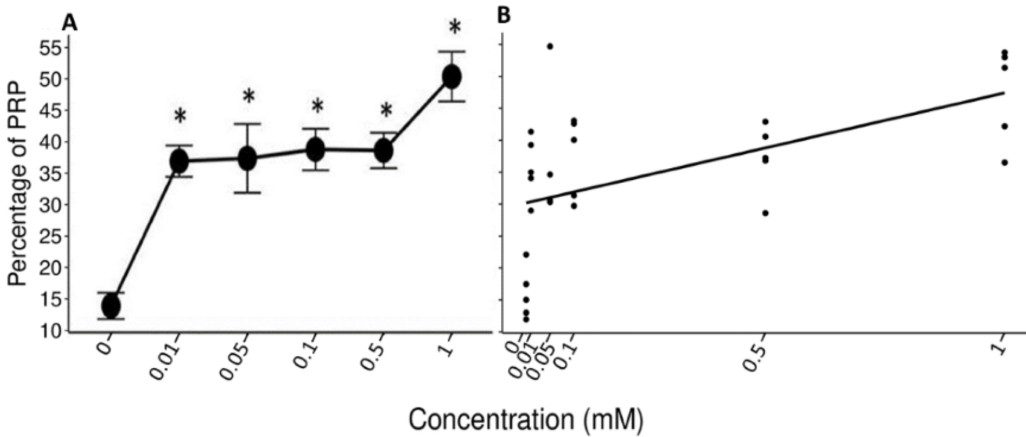

**Figure 5  Variation of %PRPs to different concentrations of tannic acid.** (A) Mean values ± S.E of the %PRP as a function of the solutions with different concentrations of tannic acid. The asterisks indicate the solutions that showed significant differences with respect to the samples of the experimental group when given the control solution. (B) Correlation between the consumed solutions with different concentrations of tannic acid and the %PRPs in the saliva of the spider monkeys. The %PRP values obtained with the concentrations of tannic acid are found on a real scale, so the data for the 0 and 0.01 mM concentrations are very close to each other on the graph.

(0.1, 0.5 and 1 mM). This result was confirmed by a positive correlation and a significant effect of the LMM, so consumption of higher concentrations of tannic acid produces more alkaline saliva. The $r^2$ value indicated that the consumption of tannic acid explains approximately 18% of the changes in the pH of the saliva, which is a very modest change, this could be related to the need of the animals to maintain stable levels of pH, to maintain a healthy oral cavity (*Llena-Puy, 2006*; *Lynge Pedersen & Belstrøm, 2019*).

An alkaline pH in the saliva of spider monkeys may be related to the consumption of fruits in the diet, since these contain acid compounds that play in plants a role in different metabolic pathways (*Batista-Silva et al., 2018*). Thus, alkaline saliva in the spider monkeys may neutralize the acidic pH of the fruits, which would help to reduce the carcinogenic potential of acids from the fruits and could contribute to the protection of oral tissues (*Llena-Puy, 2006*). Future studies should propose ways to control external sources of variation of the pH to test if this characteristic needs to maintain stable levels as previous studies indicate (*Lynge Pedersen & Belstrøm, 2019*).

## Total protein (TP)

TP concentration has been shown to depend on the type of diet that an animal consumes (*Dawes & Shaw, 1965*; *Morzel et al., 2012*), and also on the taste sensations perceived in the mouth (*Neyraud et al., 2006*; *Quintana et al., 2009*; *Torregrossa et al., 2014*). Our study is the first report of the TP concentration of the saliva for *A. geoffroyi*. The results indicate that the baseline TP concentration of the saliva of the spider monkey is lower compared to other Neotropical primates such as black howler monkeys *Alouatta pigra*, whose reported mean TP concentration was 0.8 mg/ml (*Espinosa-Gómez et al., 2018*), while the spider monkeys' mean TP concentration was 0.25 ± 0.05 mg/ml.

Previous studies suggest that the production and concentration of salivary proteins are modulated primarily by the diet, in particular by fiber content (*Neyraud et al., 2006*; *Quintana et al., 2009*; *Canon et al., 2010*). Folivorous diets are generally associated with higher fiber and tannin content (*e.g.*, howler monkey diet; *Milton, 1978*; *Dias & Rangel-Negrín, 2015*), so this may be related to higher concentrations of TP compared to frugivorous diets that generally have lower fiber and tannin content (*Klein & Klein, 1977*; *González-Zamora et al., 2009*; *Masi et al., 2015*).

The production of salivary proteins in spider monkeys is likely to be an adaptive response to their diet, maintaining a suitable environment for the processing of low-fiber foods. Recent studies have shown that the mean concentration of proteins in human saliva (1.5 mg/ml) is lower than that reported in chimpanzees, gorillas, and macaques (range approx. 3 to 5 mg/ml; *Thamadilok et al., 2020*). The authors argue that the concentrations of proteins in the saliva cannot necessarily be attributed to the phylogenetic closeness of the species, as chimpanzees showed concentrations more similar to macaques and gorillas, and not to humans, to whom they are more closely related. Similar to what was reported with humans and chimpanzees, spider monkeys showed a TP concentration lower ($0.25 \pm 0.05$ to $0.40 \pm 0.04$ mg/ml) than that reported in black howler monkeys (0.8 mg/ml), so the differences may also be associated, as in humans, to a diet with low fiber content (*Thamadilok et al., 2020*). Further studies in spider monkeys and comparisons with other Neotropical primate species are needed to corroborate this notion.

Since the $r^2$ values indicate that the consumption of tannic acid explains just a part of the TP concentration, it is important to consider other nutritional and non-nutritional elements that comprise the diet of spider monkeys that may affect this concentration. The proteins of the saliva are involved in several functions such as the perception of flavors and sensations associated with food, digestion, defense limiting or inhibiting pathogenic agents (*e.g.*, antifungal actions), and maintaining the mineralization of teeth (*Dawes et al., 2015*; *Thamadilok et al., 2020*; *Espinosa-Gómez et al., 2020*). These sources should be considered in future research projects to have a better idea of the mechanisms underlying the production of salivary proteins.

## Proline-rich proteins (PRPs)

PRPs have been reported previously in the saliva of black howler monkeys (*Espinosa-Gómez et al., 2018*), and based on the close phylogenetic relatedness of howler monkeys and spider monkeys (*De Lima et al., 2007*; *Matsushita et al., 2014*), it was suspected that these proteins might be present in spider monkey saliva as part of a mechanism to cope with tannins. We found an increase in PRPs concentrations as the concentration of ingested tannic acid increased, while the control group only showed slight variations throughout the experimental period. The variations in the control group may have been due to variations in the fruits and vegetables consumed by each individual as part of their daily diet since we did not control the daily diet of the monkeys during the study. It is important to note that the mean of %PRPs obtained from the control group was always lower than the mean of %PRPs of the experimental group. This difference indicates that the PRP concentration changes were due to the consumption of tannic acid.

A significant positive correlation and the LMM both indicate that the variations in the % PRPs were related to the consumption of tannic acid, with only a small variation of the PRPs produced between individuals. The $r^2$ value indicates that tannic acid consumption can explain approximately 48% of the changes in the value of the percentage of PRPs, therefore, suggesting that spider monkeys can modulate the PRPs percentage of saliva in response to the concentration of tannic acid. Our results provide the first evidence of the presence of PRPs in a Neotropical primate with a primarily frugivorous diet; however, a confirmation of the presence of PRPs in the saliva of spider monkeys can only be fully confirmed by an analysis of the salivary proteome, as has been done with other primates (*Thamadilok et al., 2020*; *Espinosa-Gómez et al., 2020*).

The presence of PRPs in the saliva of the spider monkeys and their variations associated with the consumption of tannic acid may be an effective defense mechanism to avoid the negative effects of consuming tannic acid since the PRPs in saliva may interact with the tannins forming tannin-protein complexes (*Pascal et al., 2007*; *Guichard et al., 2017*). This defense mechanism against tannins has been reported in other species of non-human primates (*Oppenheim, Kousvelari & Troxler, 1979*; *Schlesinger, Hay & Levine, 1989*; *Mau et al., 2011*; *Espinosa Gómez et al., 2015*; *Espinosa-Gómez et al., 2018*) and could therefore represent an adaptive physiological mechanism among primates.

The mean life span of the PRPs is 50.3 ± 24.8 h, which is longer than that of Histatins (7.2 ± 5.5 h; *Campese et al., 2009*). Due to the fact that the production of this type of protein is costly for the animals, it seems reasonable to assume that the primates produce only a limited quantity under baseline conditions, with the ability to increase PRP concentrations as a response to the consumption of unripe fruits, leaves and seeds in the diet. This may, at least partially, explain how spider monkeys adapt to limited food availability; however, they still need to include ripe fruits in their diet to survive (*Reddy et al., 1985*; *McRae & Kennedy, 2011*).

When the proportion of available ripe fruits decreases spider monkeys increase the consumption of other plant parts in their diet (*Chapman & Chapman, 1990*; *Wallace, 2005*; *Felton et al., 2008*; *Felton et al., 2009*). Under such circumstances, they are exposed to a higher concentration of tannins and organic acids, which induce physicochemical changes like an increase in the pH of the saliva to prevent damage to the oral cavity by organic acids, an increase in the TP concentration and an increase in the %PRPs in saliva to prevent tannins from precipitating digestive proteins and to reduce the bitter taste and astringent sensation caused by this compound. Based on our data, we propose a model in which frugivorous primates present physicochemical characteristics which allow them not only to feed on the plant parts they most prefer, such as ripe fruits, but also on immature fruits, which exposes them to a higher concentration of secondary metabolites such as tannic acid. They achieve this dietary flexibility by modulating salivary physicochemical characteristics (pH, PT and %PRPs) in order to feed on these plant parts without being affected by the negative effects associated with these compounds (Fig. 6).

One limitation that we had in our work was the amount of spider monkeys' saliva that we obtained, since the average was 0.213 ml per individual pool, this prevented us from doing complementary tests for this work, such as the identification of the ions responsible

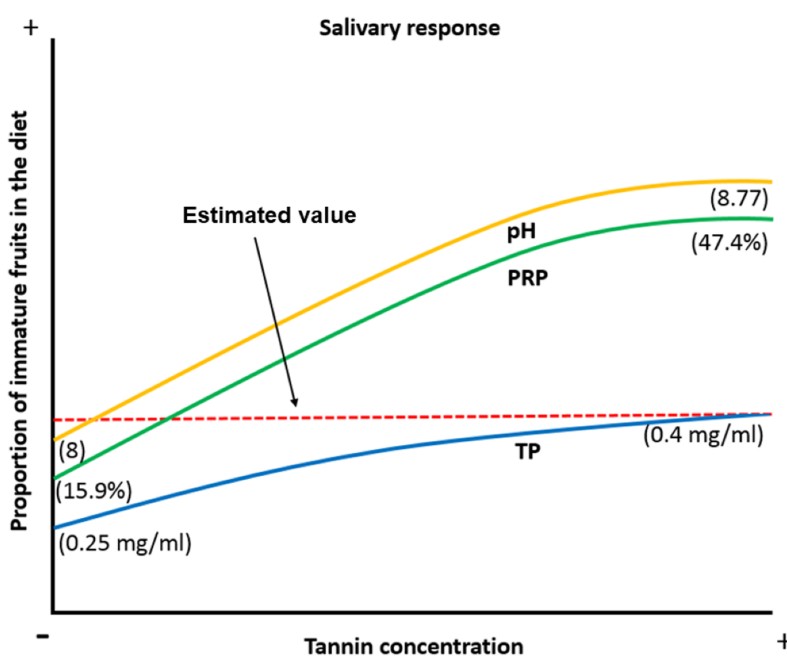

**Figure 6 Biological model of the response of the physicochemical characteristics of saliva in spider monkeys to tannin consumption.** Physicochemical characteristics of saliva in spider monkeys such as acidity-alkalinity (pH), total protein concentration (TP) and percentage of proline-rich proteins (%PRP) show changes related to the proportion of ripe and unripe fruits in their diet. The higher the proportion of unripe fruits in the diet, the more likely it is that they will be exposed to higher concentrations of tannins and acids and an increase in pH, PT and %PRP. According to our model, the estimated value is where it is considered that we obtained our maximum value that the curve reached in the production of PT when consuming immature fruits.

for the buffer capacity in saliva. Therefore, for future work, some techniques should be implemented to obtain a greater quantity of saliva.

## CONCLUSION

Our results provide the first evidence of the presence of PRPs in a Neotropical primate with a mainly frugivorous diet such as spider monkeys (*Ateles geoffroyi*); however, the presence of PRPs in the saliva of these primates can only be fully confirmed by an analysis of the salivary proteome. Spider monkeys are able to adjust the acidity-alkalinity of saliva, the concentration of total protein, and significantly increase the secretion of proline-rich proteins as a possible defense mechanism to reduce the negative effects of tannic acid in the diet and are thus able to feed on plant parts with the presence of this tannin.

## ACKNOWLEDGEMENTS

The authors thank Mr. Gildardo Castañeda for his help during the field work. A big thank you to all the monkeys who participated in our experiments; without their cooperation, nothing would be possible.

### Funding
This work was supported by Consejo Nacional de Ciencia y Tecnología CONACYT (726258). The funders had no role in study design, data collection and analysis, decision to publish, or preparation of the manuscript.

### Grant Disclosures
The following grant information was disclosed by the authors:
Consejo Nacional de Ciencia y Tecnología CONACYT: 726258.

### Competing Interests
The authors declare that they have no competing interests.

### Author Contributions
- Carlos Eduardo Ramírez-Torres conceived and designed the experiments, performed the experiments, analyzed the data, prepared figures and/or tables, authored or reviewed drafts of the article, and approved the final draft.
- Fabiola Carolina Espinosa-Gómez conceived and designed the experiments, performed the experiments, analyzed the data, authored or reviewed drafts of the article, and approved the final draft.
- Jorge E. Morales-Mávil conceived and designed the experiments, analyzed the data, authored or reviewed drafts of the article, and approved the final draft.
- J. Eduardo Reynoso-Cruz conceived and designed the experiments, performed the experiments, analyzed the data, prepared figures and/or tables, authored or reviewed drafts of the article, and approved the final draft.
- Matthias Laska conceived and designed the experiments, analyzed the data, authored or reviewed drafts of the article, and approved the final draft.
- Laura Teresa Hernández-Salazar conceived and designed the experiments, performed the experiments, analyzed the data, authored or reviewed drafts of the article, and approved the final draft.

### Animal Ethics
The following information was supplied relating to ethical approvals (*i.e.*, approving body and any reference numbers):

The Ethics Council of the Ministry of Environment and Natural Resources approved the study (SEMARNAT, 09/GS-2132/05/10).

### Field Study Permissions
The following information was supplied relating to field study approvals (*i.e.*, approving body and any reference numbers):

Field experiments were approved by the Instituto de Neuroetología (09/GS-2132/05/10).

## Data Availability

The data and code are available at figshare:

Ramírez Torres, Carlos Eduardo; Espinoza Gomez, Fabio Carolina; Morales Mavil, Jorge Eufrates; Reynoso Cruz, José Eduardo; Laska, Matthias; Hernández-Salazar, Laura (2022): Data analysis code of spider monkeys' saliva. figshare. Dataset. https://doi.org/10.6084/m9.figshare.20640756.v3.

Ramírez Torres, Carlos Eduardo; Espinoza Gomez, Fabio Carolina; Morales Mavil, Jorge Eufrates; Reynoso Cruz, José Eduardo; Laska, Matthias; Hernández-Salazar, Laura (2022): Data base PRPs in spider monkeys. figshare. Dataset. https://doi.org/10.6084/m9.figshare.20640612.v2.

## Supplemental Information

Supplemental information for this article can be found online at http://dx.doi.org/10.7717/peerj.14402#supplemental-information.

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
