# Peer review of "Influence of tannic acid concentration on the physicochemical characteristics of saliva of spider monkeys (Ateles geoffroyi)"

_PeerJ, doi:10.7717/peerj.14402_

## Round 0.1 · original submission · Minor Revisions

Dear authors

Thank you for your submission. Although the manuscript is well-written in a simple and clear way to be understood, it requires a number of Minor Revisions. Thank you for the privilege of getting a first peek at this work before it is out in the wide world and for entrusting me to comment on it before publication. I hope you find my comments and suggestions useful.

·

Basic reporting

The manuscript is well-written and easy to understand. It contains sufficient background and context.

I have a few minor suggested edits for the authors.
The terms total protein, tannin-binding salivary proteins, proline-rich proteins, and histatins should not be capitalized.
Line 114 human – should be humane
Line 139 – remove clearly
Line 144 change Saliva extraction to Saliva collection
Line 148 put end of parenthesis after USA.
Line 201 change “with” to “of”
Lines 241-244 are repeated again lines 249-252. State those results only once.
Lines 259-263 describe statistical analyses and should be in the statistical analysis section of methods.
Line 345-346 Change “since this was maintained and not altered during our study.” to “since we did not control the daily diet of the monkeys during the study.”
In the results section I think that reporting the correlations first between the tannic acid solutions and pH, TPs and PRPs and then the differences between each treatment would flow better.

I also have some suggested edits for the figures.
I suggest combining figures 2 and 3 into one figure, figures 4 and 5 into one figure, and 7 and 8 into one figure, all showing means and SE’s like in figs 2,4, and 7 and then putting the correlation lines from figs 3,5, and 8 in instead of the lines connecting the circles. Or just put each individual's data for all of the tannic acid concentrations into figures 3,5 and 8 and remove figs 2,4, and 7.
Figure 6: Put the concentration of the tannic acid solutions only over the experimental group's lanes since the control group always got the control solution. Also, the PRPs in C-F do seem to have a pinkish hue about them but I am not sure if that is true based on the image quality. You destained in acetic acid so that PRPs would destain pink/violet but never mentioned color changes in results or discussion.
Figure 9 – I like this figure!

Experimental design

The research is within the aims and scope of the journal and the research question is well defined and meaningful and fills a knowledge gap. The authors performed the experiment according to all required ethical standards for work with non-human primates.

I do have a few suggestions for details that need to be added to the methods.

How much of each solution was offered to the monkeys? Was the amount they drank recorded or did they always drink the entire amount offered? Why did you choose 8 days for each solution and then 5 days in between? Why did you have a control group that was always given just sucrose? As far as I can tell you don’t actually use their data except for %PRPs? Did their pHs or TPs change at all? I assume no, but that data should be in the results.

Line 140-141 – “Animals have a period of non-experimental, and during this time the individuals received only 30mM sucrose.” Do you mean in between the experimental replicates all animals received only 30mM sucrose?

Line 181 Change run buffer to running buffer. Why didn’t you use a sample buffer? Did you heat your samples to denature them first? If you didn’t heat your samples 1st then your molecular weights are not accurate.

Validity of the findings

The results of the study are interesting and thoroughly discussed. The findings are placed into the context of other studies, and potential problems and next steps are well laid out in the discussion.

I do have the following comments about the discussion though.

Lines 286-293 I don’t understand why the role of organic acids in metabolic pathways has anything to do with how alkaline the monkey’s saliva is, so I recommend removing reference to it. I do see why eating a diet higher in acid would need to be neutralized by alkaline saliva to protect oral tissues.

Lines 346-347: You state that the experimental group always had lower %PRPs, did you test that statistically? If yes report that in the results.

Lines 398 – 399: Don't refer to the molecular weight range of PRPs in the conclusion just state that tannic acid increases PRPs.

·

Basic reporting

Abstract
Line 22
The transition from describing previous findings of PRPs in omnivorous and folivorous primates and then describing what was assessed in spider monkeys is abrupt. Why do the authors only focus on previous findings on PRPs specifically but then say that they “therefore” assessed pH, TP concentration, and PRPs? This could easily be fixed by simply stating in the previous sentence that “variations in salivary physiochemistry as adaptations for consuming tannin-rich foods have been found in omnivorous and folivorous primates, but have not yet been reported in frugivorous species such as spider monkeys”.

Line 23
I think the instrument used to measure absorbance at 595 nm for testing TP concentration should be mentioned here, otherwise readers do not know how to interpret this statement at a glance.


Introduction
Line 39
“ripeness stage of the plant part” can be simplified to “ripeness of the plant part” and the subsequent clause can be simplified to “with ripe fruit reported to have lower concentrations compared to unripe fruit”

Lines 42-43
“changes” should be “differences” in this sentence

Line 51
Given the large number of references in this sentence, it can be difficult to follow it. I suggest changing the comma after the Dixon et al. reference to a period and then starting a new sentence beginning with “This reduced digestibility can ultimately affect the growth rate and development…”

Line 53
The second clause in this sentence can be simplified and have a redundant word removed to: “which is frequently used to assess effects…”

Line 62
The language of the second clause can be tidied up by changing it to begin with “and many have evolved…”

Line 70
Remove the unnecessary “the” here to result in “changes in salivary protein concentration…”

Line 71
End the clause ending in “...with the diet have been described” with a period and start the subsequent new sentence as “Primates that include a higher proportion…”

Line 96
Remove the comma that separates the clause ending in “..differences in precipitations” from the subsequent clause beginning with “lead to changes…”
Line 102
I suggest admitting that seasonal changes in spider monkey diets “might” require strategies to cope with the presence of plant secondary metabolites. Without additional information provided, it’s not convincing that these adaptations are necessary (i.e. “require” as stated by the authors) given that you state that spider monkey diets consist of 75-90% ripe fruit. How large are these seasonal shifts? For example, is there a part of the year when the seasonal diet substantially consists of foliage and other presumably tannin-rich plant parts?

One general comment I can make that follows from the above one is that it would be good for the authors to use an empirical example of just how drastic an effect tannin consumption can have on digestion. A recent paper by Thurau and colleagues in the American Journal of Primatology (2021, 10.1002/ajp.23239) demonstrates how high tannin intake by wild diademed sifakas can practically eliminate the protein content of the foods they ingest, even with crude protein in the diet is high. One could combine this with findings from Felton et al. 2009 (doi.org/10.1093/beheco/arp021), whom the authors elsewhere cite, to argue that even in spider monkeys dealing with tannin concentrations should theoretically play a large role in dietary intake.

Experimental design

Lines 133-139
I am not familiar with units of milli-Molar, but if I understand correctly it is specifically a measure of concentration. How much total volume was administered to the spider monkeys with each solution?

Line 147
Unsure what a “period of non-experimental” refers to here

Line 154
USA should be inside the parenthesis

Lines 169-170
Mention of saliva samples from “the three collection days” being pooled is made here. However, there is no prior description of these three collection days before. The Figure 1 legend, however, does describe how saliva was collected from each monkey on days 6-8 during each of the eight-day sampling periods. I suggest that this also be described in the body of the text of the methods section.

Line 175
The end of the sentence at the beginning of this line should end with “until further analysis”

Line 179
The authors should include a brief description of the of the Bradford. At the very least, briefly state the apparatus used so that the 595 nm wavelength can be understood in context. Does this involve use of some kind of spectrometer?

Lines 209-211
This is a suggestion more than a necessary change, but an image of the electrophoresis gel and bands being analyzed in ImajeJ would be a useful means of helping readers visualize and understand this analysis. The authors could easily just use one or two images from the larger panel of six in Figure 6.

Validity of the findings

Lines 236-238
See my comments for Figure 2 below

Lines 244-245
The text indicates that a significant difference in TP was only found between the solution of 0.5 mM tannic acid and the control solution, but asterisks are present above all the tannic acid concentrations except .05. What do these asterisks mean? No description of the asterisks is provided in the figure legend either.

Lines 245-247
The Pearson correlation coefficient is likely being influenced by the outlying samples with low TP concentration in the control samples. Given that several of these control samples have TP concentrations on par with those at the 0.5 and 1mM tannic acid concentration level, this statistically significant, albeit weak as admitted by the authors, correlation doesn’t really carry much biological meaning.

Lines 254-258
I am greatly intrigued by the substantial rise in %PRPs during the sixth sampling period of the control group. While the rise in the experimental group is still absolutely larger, do the authors have any idea as to why the control group saw a noticeable rise in their %PRPs?

Lines 255 and 263
At first, the mean %PRPs of the experimental group at the sixth tannic acid administration is stated to be 47.34% while on line 263 it is stated to be 47.35%

Lines 263-266
Similar to my comments about Figure 4 and lines 244-245 above, the text here states that a significant difference in %PRPs was only found between the control and 1 mM tannic acid samples, but asterisks are present above all the tannic acid concentrations. Moreover, %PRPs clearly tripled at the 0.01 mM tannic acid level and remained so until increasing yet again at the 1 mM level. There’s a conflict between the results of this analysis as described in the text and as presented in Figure 4.

Lines 272-275
The text describing the mechanics of the LMMs, such as which variables where included as fixed effects and which as random effects, should be moved to the Methods section. Also, on line 275, I suggest changing the text to: “the IDs of the individual monkeys in the study…”

Line 276
I believe should be corrected to say “...indicated a significant effect of concentration of tannic acid…”? Similar changes should be made the description of the results for the LMMs on TP and %PRPs. Additionally, I urge the authors to qualify their findings with regards to pH here. The effect seems to be modest, both from the standpoint of overall effect size and the low R2 and, as I pointed out above, might be influenced by a few outlying control samples.

Lines 310-312
See my comments for line 276 above. Additionally, perhaps the authors could provide a simple illustrative example as to how large a difference in pH from 8 to 9 (the difference one can eyeball in Figure 3) is in a biological sense? Readers might have difficulty immediately grasping whether, how this is a significant increase in how alkaline the saliva is, and what this means biologically.

Lines 329-341
This is very useful background information that provides important context to the author’s findings with regards to salivary TP levels in their subjects. I would urge the authors to provide similar elaboration for their findings on salivary pH levels.

Lines 358-260
Are there plant parts in the normal diets of the spider monkeys that could have induced the large increase in %PRPs at the sixth sampling period for the control group as I expressed interest in above? This could make some readers suspect as to how large an influence these background diet factors may have played in the results presented by the authors, especially given their low sample sizes. The effects that the authors detected consistently point in one direction however (i.e. higher tannic acid concentrations leading to higher %PRPs). So it may just be that the true effect size might have been smaller or larger, although still higher in the experimental group, had the authors been able to sample additional monkeys

Lines 390-391
What are the other plant parts that spider monkeys seasonally consume when ripe fruit is not as available? Is there data available on how large a proportion of the seasonal diet these plant parts make up? These data would really drive home how important these potential salivary physicochemical adaptations might be.


Figures

Figure 1
I love this figure! It’s very well done, and clearly illustrates the experimental design in a simple, easy-to-understand format. I believe “desing” is a typo for “design” in the figure legend.

Figure 3
Are their several points at the 0.5 mM concentration that are superimposed on each other due to having very similar pH values or do the authors only have two data points at this tannin concentration level? If the former, I would suggest jittering the points in R so that it is clear to viewers that there are more data points than there appear to be or at the very least stating sample sizes in the figure legend. If that latter, then the authors should make more clear how many pH samples were taken from the spider monkeys in the methods.

Figure 8
My comments about superimposed points for Figure 3 similarly apply here.

Figure 9
A fascinating and well-illustrated figure. However, it is unclear to me how the authors came up with the theoretical values of the proportion of immature fruits in the diet and what the “limit value” of TP means in this context. In all likelihood, this could be easily described in the figure legend.

Additional comments

This is a well-written manuscript that simply and clearly describes experimental results from a study that makes a worthwhile contribution to the role plant secondary metabolites play in primate dietary ecology. I applaud the authors for taking advantage of a unique study population with which they could effectively implement their experimental design. I found the paper easy, pleasant, and stimulating to read, which is not an easy thing to accomplish in my experience reading scientific literature! What follows below are my line-by-line comments on the manuscript. The overwhelming majority of them are minor, and with similarly minor revisions I would happily recommend this paper for publication. Thank you for the privilege of getting a first peek at this work before it is out in the wide world and for entrusting me to comment on it before publication. I hope you find my comments and suggestions useful.

---

## Round 0.2 · accepted · Accept

Dear authors

Thank you for addressing all of the reviewers' comments. I reviewed the manuscript myself and noticed that all comments previously done by reviewers were followed.

Your manuscript is now ready for publication in the Open Access journal PeerJ.

Congratulations.